ecology, behaviour

animals, environment, lactation, life-history, mammals, pinnipeds

**Author for correspondence:**
Rachel R. Holser
e-mail: rholser@ucsc.edu

### PUBLISHING

# Density-dependent effects on reproductive output in a capital breeding carnivore, the northern elephant seal (*Mirounga angustirostris*)

Rachel R. Holser[1], Daniel E. Crocker[2], Patrick W. Robinson[1], Richard Condit[3,4] and Daniel P. Costa[1]

[1]Department of Ecology and Evolutionary Biology, University of California Santa Cruz, 130 McAllister Way, Santa Cruz, CA 95060, USA
[2]Department of Biology, Sonoma State University, Rohnert Park, CA 94928, USA
[3]Field Museum of Natural History, 1400 South Lake Shore Drive, Chicago, IL 60605, USA
[4]Morton Arboretum, 4100 Illinois Route 53, Lisle, IL 60532, USA

RRH, 0000-0002-8668-3839; DEC, 0000-0002-7940-8011; PWR, 0000-0003-3957-8347; RC, 0000-0003-4191-1495; DPC, 0000-0002-0233-5782

All organisms face resource limitations that will ultimately restrict population growth, but the controlling mechanisms vary across ecosystems, taxa, and reproductive strategies. Using four decades of data, we examine how variation in the environment and population density affect reproductive outcomes in a capital-breeding carnivore, the northern elephant seal (*Mirounga angustirostris*). This species provides a unique opportunity to examine the relative importance of resource acquisition and density-dependence on breeding success. Capital breeders accrue resources over large temporal and spatial scales for use during an abbreviated reproductive period. This strategy may have evolved, in part, to confer resilience to short-term environmental variability. We observed density-dependent effects on weaning mass, and maternal age (experience) was more important than oceanographic conditions or maternal mass in determining offspring weaning mass. Together these findings show that the mechanisms controlling reproductive output are conserved across terrestrial and marine systems and vary with population dynamics, an important consideration when assessing the effect of extrinsic changes, such as climate change, on a population.

## 1. Introduction

Density-dependent feedback on population growth is one of the most critical ecological controls on a species [1]. While all organisms face some form of resource limitation that ultimately restricts population growth, the controlling mechanisms vary among ecosystems, taxa, and reproductive strategies [1–4]. Many non-threatened mammalian populations exist at or near carrying capacity and are limited through competition for food and habitat resources [2,5–7]. Population size, resource availability, and maternal traits all affect reproductive success, but the interaction between population density and environmental variability is understudied, particularly for wide-ranging species that separate food resources from their reproductive sites [6,8–13]. Animals that forage over large temporal and spatial scales are more buffered against food limitation, but their synchronous, aggregated breeding system creates other mechanisms by which density can act to limit reproduction [14,15]. Long-term datasets of expanding populations are critical to examining density dependent feedback, how density interacts with environmental variation, and mechanisms other than food competition that limit reproduction as populations increase. This study disentangles the effects of environmental variation and colony size on reproductive success in a large carnivore.

Elephant seals (*Mirounga* sp.) are excellent systems for investigating the dynamics of population growth, environmental variation, and reproduction. They are colonial, highly polygynous, sexually dimorphic, marine mammals that give birth to and nurse a single pup each year while fasting on land [16]. The northern species (*Mirounga angustirostris*) has been recovering from near extinction since the early 1900s [17]. The colony at Año Nuevo State Park, CA, has been studied continuously since its inception nearly 60 years ago, documenting the population as it grew rapidly, peaked, and plateaued after 2002 [17]. Prior to parturition, female northern elephant seals spend eight months foraging in the mesopelagic northeast Pacific Ocean, accumulating body stores to sustain gestation and lactation [18]. They produce high energy content milk while fasting over a 27-day lactation period [19], allowing their pup to rapidly put on mass until they are abruptly weaned. Higher weaning mass increases a pup's chance of survival, as they must rely on their body stores to sustain them for weeks between weaning and departing the colony to attempt foraging for themselves [20–24]. Colony density and maternal age are both important factors in northern elephant seal reproductive output: older females have greater success at weaning pups than younger animals, and the difference between age groups is exaggerated in high density breeding aggregations [25]. Several pinniped species exhibit sexual dimorphism and a polygamous reproductive strategy. Consequently, male pups in those species often receive greater maternal investment, particularly when resources are abundant [26–30].

In addition to demographic data and weaning masses, adult female body condition measurements have been collected since the 1990s, adding information on female foraging success and the effect of varying oceanographic conditions [18,31,32]. This time series includes multiple El Niño Southern Oscillation (ENSO) events and the marine heatwave of 2014–2015. This marine heatwave was characterized by anomalously warm sea surface temperatures that developed in the northeast Pacific Ocean in late 2013 and persisted through to 2015, causing ecosystem-level disturbances [33,34]. This dataset is uniquely positioned to investigate the influences of environmental variability and colony density on reproduction in a rapidly growing population.

We use weaning mass data from 1984–2020 to explore the effects of intrinsic (i.e. maternal traits and offspring traits) and extrinsic (i.e. environmental variability and colony density) factors on reproductive success of females. We hypothesize that: (i) offspring quality (e.g. weaning mass) will exhibit density dependence: as the population of the reproductive colony increases, quality will decrease; (ii) older females will cope better with a high-density colony; (iii) sex allocation of resources will be male biased in years with lower colony density and/or more abundant resources; and (iv) wean mass is reduced during El Niño events. Answers to these questions provide insight into the fundamental principles of population dynamics that enhance our understanding of how species will respond to changing conditions.

## 2. Methods

### (a) Study site
We conducted this study at the mainland northern elephant seal colony at Año Nuevo State Park, San Mateo County, California, USA. The colony extends across two miles of beach but breeding consistently takes place in concentrated areas where there is little or no delineation between harems (electronic supplementary material, S1 and S2). The population increased until 2002 [35] then plateaued, despite continued growth at the species level [17], suggesting that this colony has reached carrying capacity.

### (b) Adult female mass and pup mass
Elephant seals come to shore for extended fasting periods during both breeding (January–February) and moulting (April–June), allowing us to access them for monitoring and sampling. Between these haulout periods, the animals are at sea continuously for approximately three months (post-breeding) and approximately eight months (post-moult). Adult female mass gain during the gestational (post-moult) foraging trip was measured in approximately 20 individuals per year from 2004 to 2019. Females were sedated a few days before departure at the end of the moulting fast and again upon return to shore during the breeding season, following standard protocols [18]. These individuals were equipped with time-depth recorders (TDRs) and satellite transmitters (Wildlife Computers, Redmond, WA, USA or Sea Mammal Research Unit, St Andrews, UK) that provided location information through the Argos network. At each handling, the animal was weighed in a canvas sling suspended from a hanging scale with a precision of ±1 kg. During the breeding season, instruments were recovered four to seven days following birth. During these procedures, the female's pup was also weighed and flipper tagged. Additionally, adult females were sedated for physiological studies in 1991, 1992, 1995–1997, 2001–2006 and 2009. These procedures provided additional data on female arrival mass, pup birth mass, and pup weaning mass.

Measured masses are not always taken at the same time in the animal's life cycle. Consequently, all measured masses were corrected to the same life stage, accounting for variation in time spent fasting and nursing (see the electronic supplementary material, S4 for correction equations). The measured mass of the pup was corrected to birth mass based on an average mass gain of 2.2 kg d$^{-1}$ during the first few days of lactation [36]. In adult females, days spent fasting without lactating (prior to parturition) ($df$) were assumed to cost 3.0 kg d$^{-1}$ [37], while lactation days ($dl$) were assumed to cost the female 7.5 kg d$^{-1}$ [38]. Departure from and arrival at the colony was determined from either TDR or satellite records. Animals were resighted daily after arrival to assess whether they had given birth.

### (c) Weanling mass and sex ratio
A sample of weanling pups at Año Nuevo mainland has been weighed every year from 1978 to 2020 (except 1979, 1981 and 1983) following the methods outlined in Le Boeuf & Crocker [31]. Briefly, pups were marked with a unique identifier using hair bleach prior to weaning. They were designated as pups (with adult females) or weanlings (independent) during subsequent daily resight efforts. Weaned pups quickly move away from harems, making it possible to distinguish weanlings easily. Once weaned, animals were captured in a nylon restraint bag, flipper tagged, and weighed from a hanging scale with a precision of ±1 kg. As with pup and adult masses, weaning mass was corrected for days spent fasting between weaning and weighing (determined from the resight effort described above) using the equation shown in table 1 from Le Boeuf & Crocker [31] (see also the electronic supplementary material, S4).

Sex ratios were calculated using all flipper tagged young of the year that were born to flipper tagged females. This subset of individuals was selected to mitigate known sampling biases during weighing and tagging efforts owing to differing study objectives between years and differences in behaviour between male and female pups. Values for each year were tested against unity using a two-tailed binomial test.

**Table 1.** Results of generalized additive mixed effect models (GAMMs) explaining variation in weaning mass as of function of intrinsic and extrinsic factors. (Models 1.a-1.h are the entire dataset and included MomID as a random effect. Models 2.a-2.d are the subset which included birth mass and maternal arrival mass. Models in bold were the best fit for each set.)

| weanmass ~ | | log-likelihood | AICc | $R^2$ adj. | $n$ |
|---|---|---|---|---|---|
| intrinsic | | | | | |
| 1.a. | sex + s(MomAge) | −6220.798 | 12 453.7 | 0.656 | 1504 |
| 1.b. | s(MomAge) | −6229.672 | 12 469.4 | 0.650 | 1504 |
| 2.a. | PupBirthMass + Momage | −351.740 | 712.0 | 0.402 | 89 |
| 2.b. | PupBirthMass + MomMass | −364.370 | 737.2 | 0.205 | 89 |
| extrinsic | | | | | |
| 1.c. | s(population) + s(ENSO3) + s(NOI) | −5944.286 | 13 036.4 | 0.382 | 1504 |
| 1.d. | s(population) | −5940.837 | 13 063.7 | 0.380 | 1504 |
| 1.e. | s(population) + PDO | −5947.168 | 13 067.6 | 0.380 | 1504 |
| 2.c. | population + ENSO3 | −600.106 | 1206.4 | 0.024 | 89 |
| intrinsic + extrinsic | | | | | |
| **1.f.** | **sex + s(MomAge) + s(population) + ENSO3 + s(NOI)** | **−6188.196** | **12 400.6** | **0.668** | **1504** |
| 1.g. | sex + s(MomAge) + s(population) + PDO | −6191.448 | 12 403.0 | 0.665 | 1504 |
| 1.h. | sex + s(MomAge) + s(population) | −6194.996 | 12 406.1 | 0.664 | 1504 |
| **2.e.** | **PupBirthMass + MomAge + ENSO3** | **−346.927** | **711.7** | **0.457** | **89** |
| 2.d. | PupBirthMass + MomAge + population + ENSO3 | −345.173 | 713.2 | 0.471 | 89 |

## (d) Colony density

We used the annual number of pup births as a proxy for colony density during the breeding season. Values prior to 2011 are from Le Boeuf *et al.* [35], and subsequent data were collected using the same methods. The total number of adult females was estimated using the count of adult females at peak breeding plus the count of adult females 32 or 33 days prior to and following that date [39]. Based on previous studies, the number of births is assumed to be 97.5% of the total number of adult females present during the breeding season [39]. Females that skip breeding generally return to the colony before or after the breeding season [40], and therefore are not included in these counts. Pups were first observed on the island in 1961, and on the mainland in 1975 [35]. Most of the weanling and adult female data collected were from the mainland portion of the colony, therefore we excluded island data from these analyses. The time period included in this study was characterized by a rapid increase in the mainland population up through to 2002, after which the population levelled off or declined [35]. The number of females on the colony was assumed to reflect density because the area occupied by pupping females has changed relatively little since 1980, while the number of pupping females has increased 10-fold (see the electronic supplementary material, S1-S3).

## (e) Quantitative analysis

Statistical analyses were completed in R v. 3.6.1 [41]. Differences between years and between groups were tested using ANOVA with a Tukey's *post hoc* test or using student's *t*-tests, as appropriate. We only included years with a sample size greater than 50 for this analysis, which excluded 1978, 1980, 1982 and 1986. Total weaning mass sample size was $n = 4691$ distributed among 36 years. In addition to comparisons across years, we directly compare high-density and low-density reproductive characteristics by selecting years of comparatively low density (pup births less than 1200) with years of high density (pup births greater than 1900). Values reported are mean ± s.d. unless otherwise indicated.

Generalized additive mixed effect models (GAMMs) were used to assess the effects of various environmental and biological covariates on weaning mass using the R package *mgcv* [42]. The covariates included both intrinsic (i.e. maternal and pup traits) and extrinsic (i.e. onshore and offshore environment) factors (see the electronic supplementary material, S5 for complete list of covariates tested). Maternal age was included as a metric of maternal traits, with a fixed variance structure to account for heteroscedasticity. Some females had multiple pups represented in the dataset, so MomID was included as a random effect to control for repeat sampling. The number of pup births per year was used as a proxy for colony density. Three different indices of ocean conditions were evaluated, the Multivariate ENSO Index (Wolter 1993), the Pacific Decadal Oscillation (PDO) index [43], and the Northern Oscillation Index (NOI) [44]. These indices are based on surface ocean conditions that can significantly influence primary productivity, but the temporal sensitivity of the mesopelagic food web to surface conditions is not well understood [45], therefore we tested average annual ENSO and NOI values from one, two, and three years prior to each breeding. As the PDO is a longer timescale phenomenon, we used the average value for three years prior to the breeding season. For all models, Pearson's correlation coefficients were calculated for covariates to ensure that correlated covariates (absolute correlation value greater than 0.3) were not included together. PDO was highly correlated with the other indices tested, so was modelled separately from ENSO and NOI. To avoid overfitting, all models were restricted to five knots and model fit was assessed using restricted maximum-likelihood estimation.

Any data that violated the assumption that a single mother nursed a single pup throughout the average lactation period were removed from modelling analyses. This included weanlings above 170 kg of mass (super-weaners', nursed from multiple females) and below 80 kg (pups abandoned/separated from mother). These values represent the mean ± 2 standard deviations. If multiple weanlings were assigned to a single female in one year, they were all removed from the modelling dataset. Lastly, weanlings for whom maternal age was unknown were not included. The resulting subset of data spanned all years of the study except 2000 with $n = 1504$.

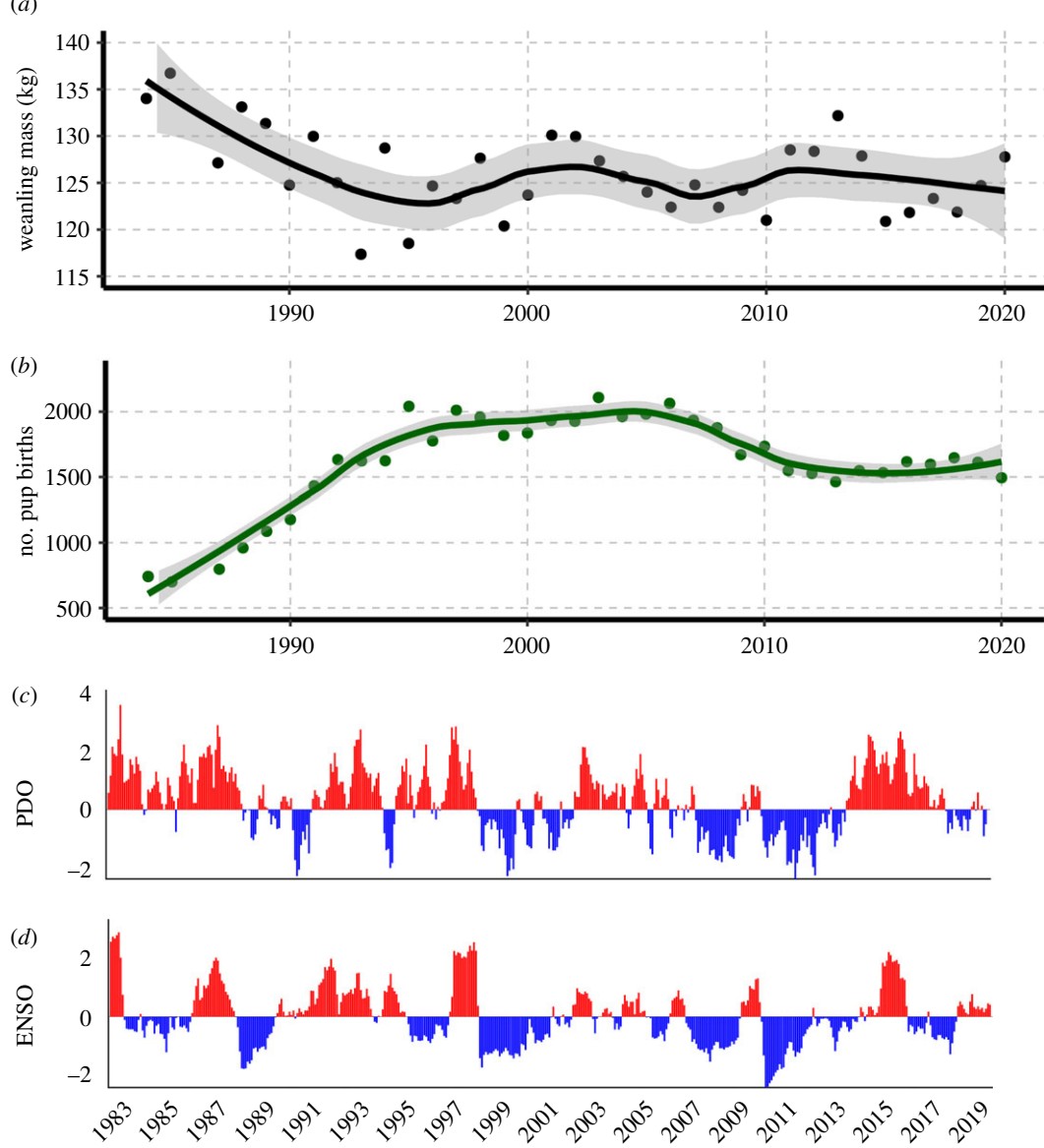

**Figure 1.** Annual mean weaning mass (*a*) and number of pup births (*b*) at the mainland portion of the Año Nuevo colony with Loess regression smoothers and shaded confidence intervals. Panels (*c*) and (*d*) show PDO and ENSO values across the sampling period. (Online version in colour.)

For a few individuals ($n = 89$) between 1991 and 2019, we measured mass at birth and maternal mass at birth, and these additional covariates were included in separate models for that subset of animals. Maternal age and arrival mass were highly correlated and were not included in models together but were tested independently. We did not have repeat sampling of mothers, so maternal identity was not included here. Additionally, we used this subset of data to quantify the relationships between maternal age, maternal size, pup birth mass, and pup weaning mass using linear models.

## 3. Results

Mean weaning mass of northern elephant seals decreased at high colony density (figure 1), and the difference was statistically significant ($131.5 \pm 25.4$ when pup births was less than 1200; $125.6 \pm 22.4$ when pup births was greater than 1900; $p = 1.967 \times 10^{-7}$; see the electronic supplementary material, S6 and S12). GAMMs showed that both intrinsic and extrinsic factors contributed to the observed variation in weaning mass (model results in table 1). Maternal age was the most important intrinsic factor and population size was the most important extrinsic driver of variation in weaning mass. At high density,

weaning mass was significantly lower across all maternal ages. Still, weaning mass increased with maternal age at both high and low density (figure 2). Ocean indices explained some of the variation in weaning mass, although they had limited explanatory power relative to other factors. Weaning mass decreased linearly with both PDO and ENSO3, while it varied nonlinearly with NOI, with weaning masses increasing for NOI values above or below $-1$ (see the electronic supplementary material, S17-S18). The best fitting models for all weanlings with known maternal age ($n = 1504$) and for weanlings with maternal mass and birth mass ($n = 89$) combined extrinsic and intrinsic factors (adj. $R^2 = 0.668$ and 0.457, respectively).

Weaning mass increased with both birth mass (electronic supplementary material, S7; $F_{1,143} = 40.0$, $p = 3.1 \times 10^{-9}$, $R^2 = 0.213$) and maternal arrival mass (electronic supplementary material, S8; $F_{1,272} = 42.33$, $p = 3.7 \times 10^{-10}$, $R^2 = 0.132$). However, older females produced heavier pups than younger females of the same mass (figure 3). While we found that larger females gave birth to larger pups (electronic supplementary material, S9; $F_{1,253} = 33.73$, $p = 1.9 \times 10^{-8}$, $R^2 = 0.114$) and weaned larger pups (electronic supplementary material, S8; also as in [32,36]), our models indicate that maternal age was a better predictor of pup mass at weaning than either maternal

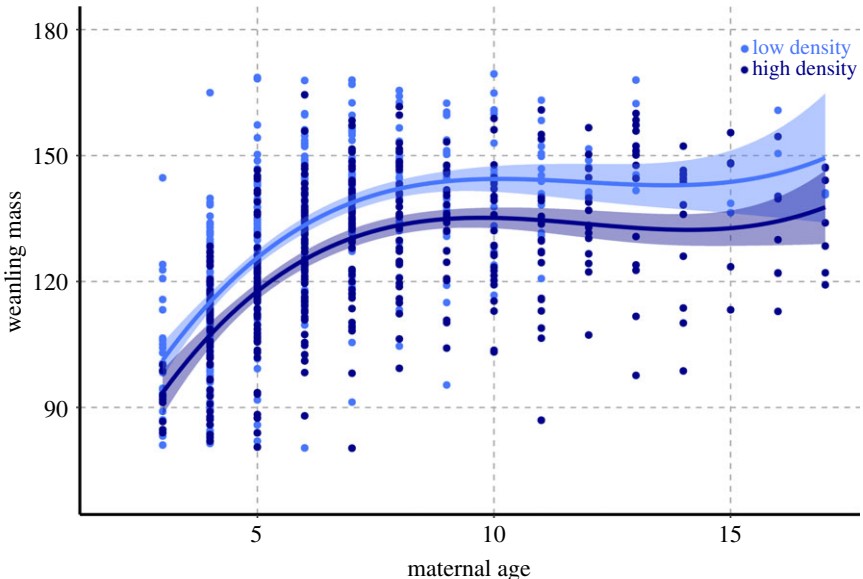

**Figure 2.** Weanling mass as a function of maternal age at low density (light blue; pup births < 1200) and high density (dark blue, pup births > 1900), with spline regression curves and shaded confidence intervals. (Online version in colour.)

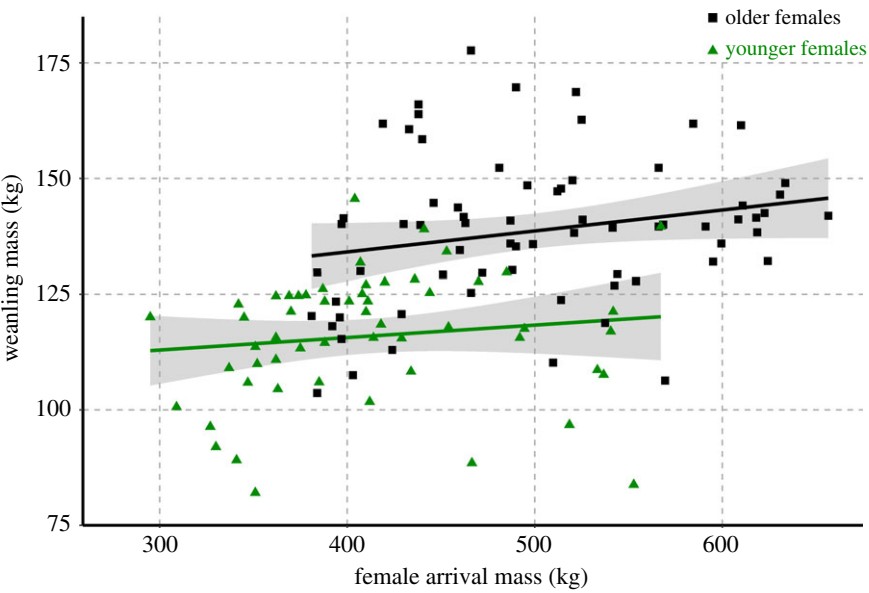

**Figure 3.** Pup wean mass as a function of female arrival mass of young (3 to 5-year-olds; green triangles) and old (9+ years of age; black squares) mothers with linear regression trendlines. (Online version in colour.)

mass or pup birth mass (electronic supplementary material, S6 and S19).

Male weanlings were, on average, heavier than female weanlings by 4.2 kg ($p = 1.6 \times 10^{-10}$). However, the magnitude of difference varied highly between years. In most years, there was no significant difference between the sexes (figure 4; electronic supplementary material, figures S6 and S12). The mass of male and female weanlings declined significantly from low to high density conditions ($-6.5$ kg $p = 5.582 \times 10^{-5}$ and $-5.3$ kg $p = 8.292 \times 10^{-4}$, respectively). The difference in mass between the sexes was lower at high density.

## 4. Discussion

We found that weaning mass declined with increasing breeding colony density. Oceanographic conditions contributed only minorly to the variation in weaning mass, compared to other factors (e.g. model 1.f. versus 1.h. in table 1). Maternal age was the strongest predictor of weaning mass, and the relationship between weaning mass and maternal age changed between high and low density (figure 2). There was a male bias in maternal resource allocation, which also declined as colony density increased.

Life-history theory suggests that offspring condition in large-bodied animals may be most affected by population density when carrying capacity is reached [2,11]. The colony at Año Nuevo transitioned from a low density, high growth rate population to a high-density population with no growth and shows clear density-dependence in offspring quality (figure 1). Weaning masses were significantly lower at high density, and the difference between the sexes decreased (electronic supplementary material, S6; figure 4). These findings are supported by a similar study conducted on southern elephant

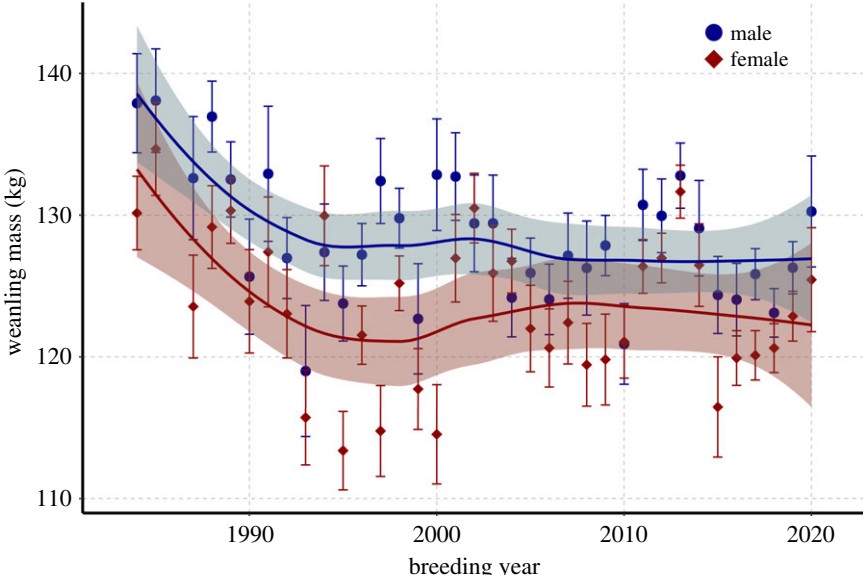

**Figure 4.** Mean weaning masses of male (blue circle) and female (red diamond) pups across years, with Loess regression curves and shaded 95% confidence intervals. (Online version in colour.)

seals (*Mirounga leonina*) on Marion Island, which also found that maternal age and population size were important drivers of weaning mass. [46] Elephant seal weaning mass integrates intrinsic factors, including maternal effects (energy stores available, milk quality, age, behaviour) and pup characteristics (behaviour, individual metabolic rate, sex), as well as extrinsic factors, such as onshore conditions (colony density, alpha male quality, beach quality, tides and storms) and at-sea foraging conditions [38,47,48]. An essential resource for reproductive success in elephant seals is the female's location on the beach. A high-quality location is above high tide and swell conditions and provides enough space for females to minimize disturbance and energy expenditure owing to interactions with conspecifics. As the colony population increases, so does overall density on the beach (see the electronic supplementary material, S2 and S3), leaving females with less space to safely nurse their pups. Previous work showed that young females have a greater reduction in reproductive output in a high-density harem compared to either more experienced individuals in the same harem or young females in a low-density harem [25].

While age and size covary in this species, size is often assumed to be the variable of import when it comes to maternal reproductive success [32]. Our findings demonstrate that age is the more important factor and should not simply be considered a proxy for size. Maternal mass captures the energetic component of reproduction, with larger females delivering more milk energy over lactation [38]. However, this effect decreases with female age, as females grow more rapidly in their early reproductive years (electronic supplementary material, S10). Maternal age represents an integration of both body condition and experience; older females have both physiological and behavioural advantages in rearing their pups [38,48,49]. Previous work shows that maternal age is 2.5 times more important to offspring growth efficiency than the energy delivered [48]. Experienced females are better at modifying pup behaviour to minimize the energy wasted through activity [48]. They are more likely to secure optimum positions within harems, reducing their energy expenditure on activities other than lactation and increasing their overall efficiency [25,38]. Furthermore, the fat content of milk produced at the start of lactation is significantly lower in young females than

that provided by prime-age females [49]. While females increase the quality of their offspring with age, some individuals may be consistently better at weaning pups throughout their lifetime [32,46,50]. We did not disentangle these effects here, as the majority of our dataset was from singly sampled adult females, but individual heterogeneity is an important question for future study.

Elephant seal weaning mass varied as a function of ocean condition indices, but maternal age and population size were more robust explanatory variables (electronic supplementary material, S6). Weaning mass in some years following unusual ocean events was statistically lower than normal (e.g. 1999 and 2015, following the 1998 El Niño and 2014 marine heatwave; table 1), while other years it was not (e.g. 1984, 2016 and 2017 following the 2015 marine heatwave and 1983 and 2016 El Niño). Previous studies reported that weaning mass declined during the warm, sardine-dominated phase of the PDO [31]. During the 1998 El Niño the rate of mass gain was lower and foraging trips were longer in adult female elephant seals during their pre-gestation foraging trip [51]. While successful reproduction is fundamentally linked to successful foraging, weaning mass is not as direct a reflection of foraging success or ocean conditions as previously thought [31,32], which is consistent with capital breeding species having reduced sensitivity to environmental disturbance compared to income breeders [14,52,53]. Capital-breeding species accrue and store resources to support reproduction over large temporal and spatial scales, whereas income breeders depend on consistent, local food resources to fuel lactation [14,15,52,54,55].

Southern elephant seals exhibit a stronger relationship between reproductive investment and ocean conditions [24,26,56–58], although maternal traits and colony conditions still show similar relationships to weaning mass as were found here [26,46]. Further comparisons between the species and between colonies within each species would provide insight into the interaction between population dynamics and the environment. Population-level indicators of poor foraging conditions may be found in other metrics, including adult female survival and frequency of skipping breeding (electronic supplementary material, S11 and results). For long-lived species, sacrificing one reproductive opportunity

to ensure future reproduction may be a greater fitness strategy than attempting to reproduce at marginal body condition, which may compromise their ability to survive, risking future reproduction [8,59–61].

Resource allocation choices can also be driven by offspring sex [7–9,62–64]. Under the model by Trivers & Willard [64], females should invest more heavily in sons than in daughters when food resources are abundant, as successful male northern elephant seals have much higher reproductive potential (up to 121 pups [65]) than the most successful female (16 pups [32]). In several ungulate species, sex bias was predicted by a combination of population density, maternal age and reproductive history, and environmental conditions, with older females exhibiting greater control over resource allocation to minimize the cost of reproduction [8,9,62]. Studies in southern elephant seals [26,66] and Antarctic fur seals (*Arctocephalus gazella*) [28,29] found similar patterns in sex-biased allocation. Although work at another northern elephant seal colony found that variation in pup sex ratio fits the Resource Competition model for sex-biased resource allocation, favouring sons in poor years [67,68], our study found that sex differences in weaning mass decreased as overall offspring quality decreased (figure 4), with no meaningful variation in offspring sex ratio (see the electronic supplementary material).

## 5. Conclusion

Our results have important implications for understanding the mechanisms controlling reproduction in capital-breeding mammals. Understanding the varying life-history patterns observed in nature and their underlying mechanisms requires long-term studies of populations, making hypothesis testing challenging, especially in large-bodied animals with slow reproductive rates and long lifespans. Time series data that have recorded changes in reproductive output as a population rapidly grows after extirpation are rare, particularly for carnivores or in marine systems (but see [69–71]). Our findings regarding density dependence and resource allocation support previous work in the field [6,8,9,11–13,63] and show that these mechanisms are conserved across terrestrial and marine mammal systems. These results reveal points of contradiction with previous studies on elephant seals (e.g. offspring sex ratio and sex bias in northern elephant seals [68,72]; the importance of ocean conditions on weaning mass in northern and southern elephant seals [26,31,57]), which illustrates the complexity of these questions and invites further investigation. The mechanisms controlling reproductive output may vary with population density, as seen here, which is an important consideration in analyses striving to assess the effect of extrinsic changes on a population. Critically, the density-dependent feedback we observed here

occurs on the reproductive colony and is independent of population density on the foraging ground.

Understanding population dynamics depends on a knowledge of vital rates and how those rates may change under varying environmental conditions. Capital breeding strategies may have evolved, in part, to confer resilience to short-term environmental variability. As a result, some species (e.g. many phocid seals) appear to avoid years of population-wide reproductive failure resulting from environmental variability seen in income breeding species within the same environment (e.g. otariids, sea birds). Individuals that attempt to breed are generally successful, even in years with poor foraging conditions. Individuals with compromised states may skip breeding and restore body condition for subsequent breeding attempts. In northern elephant seals, these traits contributed to rapid population recovery from near extinction and dramatic changes in colony density over a short period. Our findings show density-dependent changes in the mechanisms controlling reproductive success and that maternal experience and behaviour during breeding, not just body condition, is a critical determinant of effective parental investment in capital breeders.

Ethics. All animal handling was authorized under NMFS permits 496, 836, 786–1463, 87–1743, 14636, 19108, and with the approval and oversight of the University of California, Santa Cruz Institutional Animal Care and Use Committee.

Data accessibility. Data are provided in the electronic supplementary material [73] and are available from the Dryad Digital Repository: https://doi.org/10.7291/D1K099 [74] and https://doi.org/10.7291/D1D973 [75].

Authors' contributions. R.R.H.: conceptualization, data curation, formal analysis, investigation, methodology, writing—original draft, writing—review and editing; D.C.: conceptualization, methodology, writing—review and editing; R.C.: data curation, methodology, writing—review and editing; P.W.R.: data curation, funding acquisition, methodology, project administration; D.P.C.: funding acquisition, methodology, project administration, supervision, writing—review and editing. All authors gave final approval for publication and agreed to be held accountable for the work performed therein.

Competing interests. We declare we have no competing interests.

Funding. D.P.C., R.R.H. and D.E.C. were supported by the Office of Naval Research (ONR) (grant no. N00014-18-1-2822). D.E.C. was further supported by the E&P Sound and Marine Life Joint Industry Programme (JIP) of the International Association of Oil and Gas Producers (IOGP) (grant no. 00-07-23). Collection of long-term data was supported by the Tagging of Pacific Predators Program, with funds from the Gordon and Betty Moore, the Alfred P. Sloan, and the Packard Foundations.

Acknowledgements. We acknowledge all the students, volunteers, and researchers who have contributed to the Año Nuevo elephant seal dataset over the last 50+ years, especially B. Le Boeuf for pioneering the programme, J. Reiter and P. Morris for their extensive work collecting demographic data, and C. Kuhn, J. Hassrick, S. Simmons, M. Fowler, S. Peterson and L. Hückstädt for their work tracking individuals. We also thank Año Nuevo State Park and the Año Nuevo UC Natural Reserve for their ongoing support.

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
