## [Peer Review File · Proceedings of the Royal Society B: Biological Sciences]

Review History

RSPB-2020-1790.R0 (Original submission)

Review form: Reviewer 1

Recommendation

Major revision is needed (please make suggestions in comments)

Scientific importance: Is the manuscript an original and important contribution to its field?

Acceptable

General interest: Is the paper of sufficient general interest?

Acceptable

Quality of the paper: Is the overall quality of the paper suitable?

Marginal

Is the length of the paper justified?

Yes

Should the paper be seen by a specialist statistical reviewer?

No

Do you have any concerns about statistical analyses in this paper? If so, please specify them explicitly in your report.

Yes

It is a condition of publication that authors make their supporting data, code and materials available - either as supplementary material or hosted in an external repository. Please rate, if applicable, the supporting data on the following criteria.

Is it accessible?

Yes

Is it clear?

Yes

Is it adequate?

Yes

Do you have any ethical concerns with this paper?

No

Comments to the Author

I believe that density dependence is among the most important issues in population biology and the material in this paper could be important. But as presented we have not been persuaded that the results are novel and important. The authors need to review the current literature on density dependence and place their results in context. The debate regarding Sibly et al would be a fun place to begin but might not be the most relevant given the data presented here because we do not have sufficient data to rigorously evaluate the shape of the density-dependent function.

Line 37-38 This implies a concave density-dependent function (as per Fowler 1987) which seems reasonable to me, but Sibly et al. (2005, Science) challenge this premise.

Line 132 Using pups as index of density precludes reduced reproductive output as a density-dependent response. But seems to me we have adult numbers too.

Line 183 I am surprised that these differences were statistically significant. Are the +/- numbers SEs?

Line 278 "Life-history theory is at the core of ecological and population dynamics" sorry, this is not true.

Line 282, you might review data on wolf recovery from Yellowstone.

Review form: Reviewer 2

Recommendation

Major revision is needed (please make suggestions in comments)

Scientific importance: Is the manuscript an original and important contribution to its field?

Good

General interest: Is the paper of sufficient general interest?

Good

Quality of the paper: Is the overall quality of the paper suitable?

Marginal

Is the length of the paper justified?

Yes

Should the paper be seen by a specialist statistical reviewer?

No

Do you have any concerns about statistical analyses in this paper? If so, please specify them explicitly in your report.

Yes

It is a condition of publication that authors make their supporting data, code and materials available - either as supplementary material or hosted in an external repository. Please rate, if applicable, the supporting data on the following criteria.

Is it accessible?

Yes

Is it clear?

Yes

Is it adequate?

Yes

Do you have any ethical concerns with this paper?

No

Comments to the Author

1. Line 4: Word(s) seem(s) to be missing from "is primarily to ungulates"
2. Lines 32-34: I was a bit surprised that you didn't mention the probability of producing offspring.
3. Line 132: Did you have multiple functional forms for your proxy of abundance? That is, did you only consider linear relationships between abundance and the response variable(s)? I ask because one might imagine a non-linear relationship (e.g., theta-logistic like patterns).
4. Line 151: Did you constrain the flexibility of the fitted curve, e.g., using knots? If so, how? And, why or why not?
5. Lines 152-153: Did you use some threshold value for the correlation when deciding on including/excluding a variable? What was the range of values for the correlations and were any covariates eliminated? If so, how did you choose which to eliminate?
6. Analysis methods in general: (1) did you have repeat observations on individual mothers in multiple years, and, (2) if so, how did you handle that in the analyses?
7. Line 153: It would be helpful if you presented which response variables were analyzed and what competing models were evaluated for each (or note that the same models were used for each). You note that model selection was done, but it's not clear to me from the presentation what the list of competing models was or how models were constructed. Did you evaluate all-possible models or just models with different versions of the indices of ocean conditions? What of other covariates and how they were in/out of competing models? Finally, did you have any specific predictions regarding the covariates that might be useful to describe here?
8. Line 182: "higher than in the" instead of "higher than the"

9. Line 183: I don't follow the phrase that reads, " 127.5 ± 23.5 versus 125.1 ± 21.7 , respectively; $p=0.0024$ ". First, what of the biological significance of the difference: it might be helpful to consider that topic in introduction or methods. And, it would help to know what the value provided after the "+-" sign represents, e.g., if it's SD, we might want to have the SE of the mean and the estimated difference along with a CI on the difference. Further, the values after the "+-" sign are quite large relative to the estimated, which further indicates a need to. Understand what values are being provided after the "+-" sign. Finally, this paper provides a mix of significance testing (p-values presented here) and information-theoretic results (AICc differences), which is non-standard and a bit confusing. It might be helpful to pick an analysis paradigm and use it throughout.
10. Lines 193-194: Here the paper is relying on p-values it seems rather than, as stated on lines 153-154, "the Akaike Information Criterion (AICc)". Whatever approach is chosen should be justified and used consistently in my opinion.
11. Lines 192-196: Here (and throughout much of the results), it is not possible to assess the magnitude of the differences from the main text. Yet, I think that many readers will be more interested in those features of the data than in values such as the % of deviance, F statistics or P-values. It would be helpful to bring more of the biological results into the text to accompany the statistical results.
12. Line 200: By "significantly", are you referring to biological or statistical significance or both? IT would be useful to clarify and justify the choice.
13. Lines 217-220: Same comments as made above regarding text on lines 192-196.
14. Line 236: Same comment as made above regarding text on line 200.
15. Lines 253-256: Same comment as made above regarding text on line 183.
16. Results in general: I found it difficult to evaluate which of the covariates were most important in explaining variation in your response variable(s) and suggest providing more concrete information on the topic in Results.
17. Literature Cited: There are other papers on marine mammals in particular that relate to some of the topics covered in this paper and that could be added, e.g., Bowen et al. 2015. *Ecology and Evolution*: 5(7): 1412- 1424.
18. Table 1: By "Year (RE)", are you indicating that year was treated as a random effect? This should be explained here and in Methods regarding the how and why of including it this way.
19. Table 2: What values are being provided after the "+-" sign?
20. Figure 1: It appears that both response variables hit what might be considered a threshold well before the point in the study at which you consider the density to have transitioned from low to high. Is this adequately discussed? What are the implications?
21. Figures 2 and 3: Confidence bands to provide measures of uncertainty are lacking but would be useful.

Decision letter (RSPB-2020-1790.R0)

02-Sep-2020

Dear Miss Holser:

I am writing to inform you that your manuscript RSPB-2020-1790 entitled "Density-dependent effects on reproductive success in a capital breeding carnivore, the northern elephant seal (*Mirounga angustirostris*)" has, in its current form, been rejected for publication in *Proceedings B*.

This action has been taken on the advice of referees, who have recommended that substantial revisions are necessary. With this in mind we would be happy to consider a resubmission, provided the comments of the referees are fully addressed. However please note that this is not a provisional acceptance.

Sincerely,
Dr Locke Rowe
<mailto:proceedingsb@royalsociety.org>

Associate Editor
Board Member: 1
Comments to Author:

Two experts in this subject area have now reviewed your manuscript. Both reviewers thought the extensive, long-term data presented in this ms have potential for contributing to our understanding of one or more basic questions of population dynamics. But, both reviewers also felt strongly that the importance of the study is less than clear because the results/statistics were not embedded in a broader biological narrative focused on a specific issue or question in population biology. In other words, the statistical findings (significant or otherwise) were not judged to be of sufficient value because they did not directly relate back to a specific question or hole in our knowledge (i.e, the narrative). To that end, the reviewers suggested that there are many possible narratives to choose from, and that the published literature could be better

incorporated to develop a narrative. Finally, Reviewer 2 in particular thought the ms could be more strictly organized and clearer-particularly the presentation of results.

In closing, I'm sorry I don't have more positive news. Substantial work is needed to take full advantage of this rather extraordinary dataset, but with that, I think there's potential to make a significant contribution to our general understanding of populations. So, I hope you find the reviewers' comments to be helpful moving forward.

Reviewer(s)' Comments to Author:

Referee: 1

Comments to the Author(s)

I believe that density dependence is among the most important issues in population biology and the material in this paper could be important. But as presented we have not been persuaded that the results are novel and important. The authors need to review the current literature on density dependence and place their results in context. The debate regarding Sibly et al would be a fun place to begin but might not be the most relevant given the data presented here because we do not have sufficient data to rigorously evaluate the shape of the density-dependent function.

Line 37-38 This implies a concave density-dependent function (as per Fowler 1987) which seems reasonable to me, but Sibly et al. (2005, Science) challenge this premise.

Line 132 Using pups as index of density precludes reduced reproductive output as a density-dependent response. But seems to me we have adult numbers too.

Line 183 I am surprised that these differences were statistically significant. Are the +/- numbers SEs?

Line 278 "Life-history theory is at the core of ecological and population dynamics" sorry, this is not true.

Line 282, you might review data on wolf recovery from Yellowstone.

Referee: 2

Comments to the Author(s)

1. Line 4: Word(s) seem(s) to be missing from "is primarily to ungulates"

2. Lines 32-34: I was a bit surprised that you didn't mention the probability of producing offspring.

3. Line 132: Did you have multiple functional forms for your proxy of abundance? That is, did you only consider linear relationships between abundance and the response variable(s)? I ask because one might imagine a non-linear relationship (e.g., theta-logistic like patterns).

4. Line 151: Did you constrain the flexibility of the fitted curve, e.g., using knots? If so, how? And, why or why not?

5. Lines 152-153: Did you use some threshold value for the correlation when deciding on including/excluding a variable? What was the range of values for the correlations and were any covariates eliminated? If so, how did you choose which to eliminate?

6. Analysis methods in general: (1) did you have repeat observations on individual mothers in multiple years, and, (2) if so, how did you handle that in the analyses?

7. Line 153: It would be helpful if you presented which response variables were analyzed and what competing models were evaluated for each (or note that the same models were used for each). You note that model selection was done, but it's not clear to me from the presentation what the list of competing models was or how models were constructed. Did you evaluate all-possible models or just models with different versions of the indices of ocean conditions? What of other

covariates and how they were in/out of competing models? Finally, did you have any specific predictions regarding the covariates that might be useful to describe here?

8. Line 182: “higher than in the” instead of “higher than the”

9. Line 183: I don’t follow the phrase that reads, “ 127.5 ± 23.5 versus 125.1 ± 21.7 , respectively; $p=0.0024$ ”. First, what of the biological significance of the difference: it might be helpful to consider that topic in introduction or methods. And, it would help to know what the value provided after the “+–” sign represents, e.g., if it’s SD, we might want to have the SE of the mean and the estimated difference along with a CI on the difference. Further, the values after the “+–” sign are quite large relative to the estimated, which further indicates a need to. Understand what values are being provided after the “+–” sign. Finally, this paper provides a mix of significance testing (p-values presented here) and information-theoretic results (AICc differences), which is non-standard and a bit confusing. It might be helpful to pick an analysis paradigm and use it throughout.

10. Lines 193-194: Here the paper is relying on p-values it seems rather than, as stated on lines 153-154, “the Akaike Information Criterion (AICc)”. Whatever approach is chosen should be justified and used consistently in my opinion.

11. Lines 192-196: Here (and throughout much of the results), it is not possible to assess the magnitude of the differences from the main text. Yet, I think that many readers will be more interested in those features of the data than in values such as the % of deviance, F statistics or P-values. It would be helpful to bring more of the biological results into the text to accompany the statistical results.

12. Line 200: By “significantly”, are you referring to biological or statistical significance or both? IT would be useful to clarify and justify the choice.

13. Lines 217-220: Same comments as made above regarding text on lines 192-196.

14. Line 236: Same comment as made above regarding text on line 200.

15. Lines 253-256: Same comment as made above regarding text on line 183.

16. Results in general: I found it difficult to evaluate which of the covariates were most important in explaining variation in your response variable(s) and suggest providing more concrete information on the topic in Results.

17. Literature Cited: There are other papers on marine mammals in particular that relate to some of the topics covered in this paper and that could be added, e.g., Bowen et al. 2015. *Ecology and Evolution*: 5(7): 1412– 1424.

18. Table 1: By “Year (RE)”, are you indicating that year was treated as a random effect? This should be explained here and in Methods regarding the how and why of including it this way.

19. Table 2: What values are being provided after the “+–” sign?

20. Figure 1: It appears that both response variables hit what might be considered a threshold well before the point in the study at which you consider the density to have transitioned from low to high. Is this adequately discussed? What are the implications?

21. Figures 2 and 3: Confidence bands to provide measures of uncertainty are lacking but would be useful.

Author's Response to Decision Letter for (RSPB-2020-1790.R0)

See Appendix A.

RSPB-2021-1258.R0

Review form: Reviewer 3

Recommendation

Major revision is needed (please make suggestions in comments)

Scientific importance: Is the manuscript an original and important contribution to its field?

Acceptable

General interest: Is the paper of sufficient general interest?

Good

Quality of the paper: Is the overall quality of the paper suitable?

Good

Is the length of the paper justified?

Yes

Should the paper be seen by a specialist statistical reviewer?

No

Do you have any concerns about statistical analyses in this paper? If so, please specify them explicitly in your report.

No

It is a condition of publication that authors make their supporting data, code and materials available - either as supplementary material or hosted in an external repository. Please rate, if applicable, the supporting data on the following criteria.

Is it accessible?

Yes

Is it clear?

Yes

Is it adequate?

Yes

Do you have any ethical concerns with this paper?

No

Comments to the Author

General comments:

This interesting paper uses an enviable dataset of immense value, to consider drivers (intrinsic and extrinsic) of population growth on reproductive output in northern elephant seals. The paper generally focusses on the possible effect that density-dependence in an increasing population has on reproductive output variables (e.g. weaning mass). Other parameters of relevance are also

discussed, such as the age of mothers as related to weanling mass. I notice that the manuscript has undergone a previous round of reviews. The modeling approach and statistical analyses have been thoroughly reviewed in earlier rounds and appears to have been appropriately revised. My concern however lies with the underlying biological assumptions related to the scale at which density dependence is being tested; specifically how the breeding behavior of the species, especially as related to maternal age (which is central to such density dependence assessment), is treated in the methodology. I elaborate on this below.

Chiefly, while I agree with the authors that pup births should be used as the density metric, I have concerns over the scale of what is defined as 'the population' and thus the 'density' thereof. From the outset, it is clear that an increase in absolute numbers of NES pups relates strongly with a decline in weaning mass. However, this is not the same as saying an increased density of individuals on land is cause for observed decrease in weaning mass, because this is a question of scale of distribution and spacing of the individuals on land. This in turn is heavily dependent upon the behavior of the species. Specifically, how 'the population' relates to effects of density at the individual level, which would ultimately translate to weanling mass, given that age and experience of individual mothers (and thus their behavior and physical location in colonies on land) is provided as essential explanation for the density effect (Lines 207-214). The terms 'population' and 'colony' are used vaguely in the manuscript. For those not familiar with the Ano Nuevo NES assemblage it is important to clearly explain if the NES here are assembled in one large continuous population more synonymous with colony (likely not), or if as suspected, the population here is in fact a 'colony' of several harems of varying sizes dotted along the coastline. Are all harems distinct and separated from each other or are some margins vague and harems merge into each other? These questions are critical to address because I am concerned that a confounding argument of scale of study may be associated therewith. Clearly an increasing population will mean more individual animals within a defined area as a whole, but does this mean that harems are in closer proximity to each other, or perhaps merge? Does an increase in population size mean greater congestion (i.e. density) within harems? Does it mean the disappearance of small harems because all harems grow beyond a specific size threshold? Ultimately, are a larger number of NES merely spread over a larger space at a constant density or is the density of harems increased? These are vital considerations to be sure that what the authors are testing here as terrestrial population density dependence is indeed causative to the responses, i.e. changes in weaning mass. Or are increased numbers of animals in the population resulting in a density dependent effect at foraging grounds at sea (not the focus of this study). Or are we seeing a response related to how increased numbers of animals are changing the dynamics of terrestrial harems, which we know have different consequences for mothers of different ages. Clearly, density is integral but establishing at what scale the effect is would be a far more valuable addition to our knowledge than what is currently proposed in the manuscript I think. S8 & S10 seem to be valuable additions to the main manuscript rather than being deferred to the Suppl Mats. Also, the means illustrated in S10 as compared to the total population combined decline in weanling masses over time (S8), seems to me to indicate that the age distribution/composition of the adult population in harems has changed over time. Colony density as a function of population increase will be heavily dependent on available space for harems, and will likely vary between different harems? Evidence from SES suggest that young females are more regularly associated with small harems, while older more experienced females associate in larger harems with varying consequences for their pups (Pistorius et al. 2001 Polar Biol; McMahon & Bradshaw 2004 Behav Soc Biol; Postma et al. 2013 Polar Biol). If NES do the same the potential caveat here is that harems of different densities present differing female age distributions, which will confound the conclusion that density-dependence at the population level is driving the observed breeding output changes.

I feel that the manuscript is not streamlined in its present form, nor do I think it is adequately contextualized within existing literature. A host of highly relevant papers, some asking/answering several of the same questions (albeit for SES) seem to have been missed. I provide some added examples in the specific comments below, but from the outset Oosthuizen et al. (2015 - Ecosphere), seems to be a glaring omission. That paper; "Decomposing the variance in southern elephant seal weaning mass: partitioning environmental signals and maternal effects",

provides key context to many aspects addressed here (e.g. their figure 4 looks very similar to your figure 2).

Specific comments:

Title: 'Reproductive output' would be better than 'reproductive success'. "LRO is a slightly more general concept than lifetime reproductive success (Clutton-Brock 1988) because it can accommodate many different operational definitions of reproduction, as are often encountered in ecological and demographic analysis." - <https://link.springer.com/article/10.1007/s12080-017-0335-2#ref-CR18>

Line 10: This seems to be a strong statement for what appears to be fairly limited male-biased allocation of resources to offspring?

Line 10-11: Just a note - while this may be novel for NES, it is not for the southern species (see Oosthuizen et al. 2015 – Ecosphere).

Line 59: incomplete sentence.

Line 61, Hypothesis 2: Refer to general comments. Your design is not necessarily testing a higher density population, but rather a larger (in number) population, unless more detail is provided that motivates that the population has both grown and increased in density at the terrestrial site.

Line 87: if they were measured then why did they need to be corrected? Unclear. I'm assuming because measurements were not always taken at times in the life cycle needed for later analyses. State your reasoning here.

Line 139-140: see Oosthuizen et al. 2015 Ecosphere, Fig 5. Individual heterogeneity in the context of the general comments related to individual age related behaviour of female at the terrestrial breeding site may in fact be a critical component of what happened demographically to this population as it grew.

Line 144: would Chlorophyll-a be a useful variable to include?

Lines 182-185: Might, as stated, the crux in the argument of declining weaning mass over time not be with how the breeding season terrestrial behaviour of older females and/or the composition of females in different harems has changed over time?

Line 240-241: speculative, suggest deletion.

Line 242: Indeed. Several pertinent references from the work on SES at Marion Island may provide added insight on this specific topic here:

- Pistorius et al. 2001. Pup mortality in southern elephant seals at Marion Island. *Polar Biol* 24:828–831

- Postma et al. 2013. Age-related reproductive variation in a wild marine mammal population. *Polar Biology* 36:719–729.

- Oosthuizen et al. 2018. Phenotypic selection and covariation in the life-history traits of elephant seals: Heavier offspring gain a double selective advantage. *Oikos* 127:875–889

- Oosthuizen et al. 2019. Individual heterogeneity in life-history trade-offs with age at first reproduction in capital breeding elephant seals. *Population Ecology* 61:421–435.

- Oosthuizen et al. 2021. Positive early-late life-history trait correlations in elephant seals. *Ecology* 102: e03288

Line 246: 'higher' may be better replaced with 'greater'?

Line 247-248: again see above Oosthuizen et al. 2018, 2019, 2021 references.

Line 249-260: this paragraph is valuable, but feels a little 'added on' here as an afterthought almost. Attempt to link better.

Lines 286-289: based on the general comments above this line needs some reconsideration.

Decision letter (RSPB-2021-1258.R0)

19-Jul-2021

Dear Miss Holser:

Your manuscript has now been peer reviewed and the reviews have been assessed by myself. The reviewers' comments (not including confidential comments to the Editor) are included at the end of this email for your reference. As you will see, the reviewer and I have raised some concerns with your manuscript and we would like to invite you to revise your manuscript to address them.

We do not allow multiple rounds of revision so we urge you to make every effort to fully address all of the comments at this stage. If deemed necessary your manuscript will be sent back to one or more of the original reviewers for assessment. If the original reviewers are not available we may invite new reviewers. Please note that we cannot guarantee eventual acceptance of your manuscript at this stage.

Research ethics:

Use of animals and field studies:

It is a condition of publication that you make available the data and research materials supporting the results in the article (<https://royalsociety.org/journals/authors/author-guidelines/#data>). Datasets should be deposited in an appropriate publicly available repository and details of the associated accession number, link or DOI to the datasets must be included in the Data Accessibility section of the article (<https://royalsociety.org/journals/ethics-policies/data-sharing-mining/>). Reference(s) to datasets should also be included in the reference list of the article with DOIs (where available).

If you wish to submit your data to Dryad (<http://datadryad.org/>) and have not already done so you can submit your data via this link [http://datadryad.org/submit?journalID=RSPB&manu=\(Document not available\)](http://datadryad.org/submit?journalID=RSPB&manu=(Document%20not%20available)), which will take you to your unique entry in the Dryad repository.

Please submit a copy of your revised paper within three weeks. If we do not hear from you within this time your manuscript will be rejected. If you are unable to meet this deadline please let us know as soon as possible, as we may be able to grant a short extension.

Best wishes,
Dr Locke Rowe
mailto:proceedingsb@royalsociety.org

Editor

Comments to Author:

I would like to first thank the authors for their careful edits and responses to the original two reviews and the AE's comments. The initial referees and the referee of this current version all see great value in these data and the questions. I agree with this view.

The current referee would like to see more discussion of the definition of population and colony being used here. This will enable a better interpretation of the results by readers. The referee includes several additional comments that will be helpful in a revision. Among these is a suggestion to include a little bit more related literature.

I would add to this that I would like to see a few more lines justifying the hypotheses in the introduction, particularly 2 and 3.

Reviewer(s)' Comments to Author:

Referee: 3

Comments to the Author(s).

General comments:

This interesting paper uses an enviable dataset of immense value, to consider drivers (intrinsic and extrinsic) of population growth on reproductive output in northern elephant seals. The paper generally focusses on the possible effect that density-dependence in an increasing population has

on reproductive output variables (e.g. weaning mass). Other parameters of relevance are also discussed, such as the age of mothers as related to weanling mass. I notice that the manuscript has undergone a previous round of reviews. The modeling approach and statistical analyses have been thoroughly reviewed in earlier rounds and appears to have been appropriately revised. My concern however lies with the underlying biological assumptions related to the scale at which density dependence is being tested; specifically how the breeding behavior of the species, especially as related to maternal age (which is central to such density dependence assessment), is treated in the methodology. I elaborate on this below.

Chiefly, while I agree with the authors that pup births should be used as the density metric, I have concerns over the scale of what is defined as 'the population' and thus the 'density' thereof. From the outset, it is clear that an increase in absolute numbers of NES pups relates strongly with a decline in weaning mass. However, this is not the same as saying an increased density of individuals on land is cause for observed decrease in weaning mass, because this is a question of scale of distribution and spacing of the individuals on land. This in turn is heavily dependent upon the behavior of the species. Specifically, how 'the population' relates to effects of density at the individual level, which would ultimately translate to weanling mass, given that age and experience of individual mothers (and thus their behavior and physical location in colonies on land) is provided as essential explanation for the density effect (Lines 207-214). The terms 'population' and 'colony' are used vaguely in the manuscript. For those not familiar with the Ano Nuevo NES assemblage it is important to clearly explain if the NES here are assembled in one large continuous population more synonymous with colony (likely not), or if as suspected, the population here is in fact a 'colony' of several harems of varying sizes dotted along the coastline. Are all harems distinct and separated from each other or are some margins vague and harems merge into each other? These questions are critical to address because I am concerned that a confounding argument of scale of study may be associated therewith. Clearly an increasing population will mean more individual animals within a defined area as a whole, but does this mean that harems are in closer proximity to each other, or perhaps merge? Does an increase in population size mean greater congestion (i.e. density) within harems? Does it mean the disappearance of small harems because all harems grow beyond a specific size threshold? Ultimately, are a larger number of NES merely spread over a larger space at a constant density or is the density of harems increased? These are vital considerations to be sure that what the authors are testing here as terrestrial population density dependence is indeed causative to the responses, i.e. changes in weaning mass. Or are increased numbers of animals in the population resulting in a density dependent effect at foraging grounds at sea (not the focus of this study). Or are we seeing a response related to how increased numbers of animals are changing the dynamics of terrestrial harems, which we know have different consequences for mothers of different ages. Clearly, density is integral but establishing at what scale the effect is would be a far more valuable addition to our knowledge than what is currently proposed in the manuscript I think. S8 & S10 seem to be valuable additions to the main manuscript rather than being deferred to the Suppl Mats. Also, the means illustrated in S10 as compared to the total population combined decline in weanling masses over time (S8), seems to me to indicate that the age distribution/composition of the adult population in harems has changed over time. Colony density as a function of population increase will be heavily dependent on available space for harems, and will likely vary between different harems? Evidence from SES suggest that young females are more regularly associated with small harems, while older more experienced females associate in larger harems with varying consequences for their pups (Pistorius et al. 2001 Polar Biol; McMahon & Bradshaw 2004 Behav Soc Biol; Postma et al. 2013 Polar Biol). If NES do the same the potential caveat here is that harems of different densities present differing female age distributions, which will confound the conclusion that density-dependence at the population level is driving the observed breeding output changes.

I feel that the manuscript is not streamlined in its present form, nor do I think it is adequately contextualized within existing literature. A host of highly relevant papers, some asking/answering several of the same questions (albeit for SES) seem to have been missed. I provide some added examples in the specific comments below, but from the outset Oosthuizen et al. (2015 - Ecosphere), seems to be a glaring omission. That paper; "Decomposing the variance in southern elephant seal weaning mass: partitioning environmental signals and maternal effects",

provides key context to many aspects addressed here (e.g. their figure 4 looks very similar to your figure 2).

Specific comments:

Title: 'Reproductive output' would be better than 'reproductive success'. "LRO is a slightly more general concept than lifetime reproductive success (Clutton-Brock 1988) because it can accommodate many different operational definitions of reproduction, as are often encountered in ecological and demographic analysis." - <https://link.springer.com/article/10.1007/s12080-017-0335-2#ref-CR18>

Line 10: This seems to be a strong statement for what appears to be fairly limited male-biased allocation of resources to offspring?

Line 10-11: Just a note - while this may be novel for NES, it is not for the southern species (see Oosthuizen et al. 2015 - Ecosphere).

Line 59: incomplete sentence.

Line 61, Hypothesis 2: Refer to general comments. Your design is not necessarily testing a higher density population, but rather a larger (in number) population, unless more detail is provided that motivates that the population has both grown and increased in density at the terrestrial site.

Line 87: if they were measured then why did they need to be corrected? Unclear. I'm assuming because measurements were not always taken at times in the life cycle needed for later analyses. State your reasoning here.

Line 139-140: see Oosthuizen et al. 2015 Ecosphere, Fig 5. Individual heterogeneity in the context of the general comments related to individual age related behaviour of female at the terrestrial breeding site may in fact be a critical component of what happened demographically to this population as it grew.

Line 144: would Chlorophyll-a be a useful variable to include?

Lines 182-185: Might, as stated, the crux in the argument of declining weaning mass over time not be with how the breeding season terrestrial behaviour of older females and/or the composition of females in different harems has changed over time?

Line 240-241: speculative, suggest deletion.

Line 242: Indeed. Several pertinent references from the work on SES at Marion Island may provide added insight on this specific topic here:

- Pistorius et al. 2001. Pup mortality in southern elephant seals at Marion Island. *Polar Biol* 24:828–831

- Postma et al. 2013. Age-related reproductive variation in a wild marine mammal population. *Polar Biology* 36:719–729.

- Oosthuizen et al. 2018. Phenotypic selection and covariation in the life-history traits of elephant seals: Heavier offspring gain a double selective advantage. *Oikos* 127:875–889

- Oosthuizen et al. 2019. Individual heterogeneity in life-history trade-offs with age at first reproduction in capital breeding elephant seals. *Population Ecology* 61:421–435.

- Oosthuizen et al. 2021. Positive early-late life-history trait correlations in elephant seals. *Ecology* 102: e03288

Line 246: 'higher' may be better replaced with 'greater'?

Line 247-248: again see above Oosthuizen et al. 2018, 2019, 2021 references.

Line 249-260: this paragraph is valuable, but feels a little 'added on' here as an afterthought almost. Attempt to link better.

Lines 286-289: based on the general comments above this line needs some reconsideration.

Author's Response to Decision Letter for (RSPB-2021-1258.R0)

See Appendix B.

RSPB-2021-1258.R1 (Revision)

Review form: Reviewer 1

Recommendation

Accept as is

Scientific importance: Is the manuscript an original and important contribution to its field?

Excellent

General interest: Is the paper of sufficient general interest?

Excellent

Quality of the paper: Is the overall quality of the paper suitable?

Excellent

Is the length of the paper justified?

Yes

Should the paper be seen by a specialist statistical reviewer?

No

Do you have any concerns about statistical analyses in this paper? If so, please specify them explicitly in your report.

No

It is a condition of publication that authors make their supporting data, code and materials available - either as supplementary material or hosted in an external repository. Please rate, if applicable, the supporting data on the following criteria.

Is it accessible?

Yes

Is it clear?

Yes

Is it adequate?

Yes

Do you have any ethical concerns with this paper?

No

Comments to the Author

The earlier reviewer concerns have been thoroughly addressed in my opinion. I have no further comments or concerns. This manuscript is a valuable contribution to our knowledge. Well done on a fine piece of work.

Decision letter (RSPB-2021-1258.R1)

17-Sep-2021

Dear Miss Holser

I am pleased to inform you that your manuscript entitled "Density-dependent effects on reproductive output in a capital breeding carnivore, the northern elephant seal (*Mirounga angustirostris*)" has been accepted for publication in Proceedings B. Thanks for your revisions on this manuscript.

Data Accessibility section

Open Access

Your article has been estimated as being 9 pages long. Our Production Office will be able to confirm the exact length at proof stage.

Paper charges

Sincerely,

Dr Locke Rowe

Appendix A

We want to thank both reviewers and the associate editor for their thoughtful comments. Their questions prompted some detailed reworking of a portion of the methods and results sections that hopefully clarify the quantitative elements of the paper. In addition, we clarified the narrative of the paper (particularly within the introduction) to focus on disentangling the effects of density dependence and environmental variation on reproduction, which requires long-term data sets that includes large changes in density along with substantial variation in environment. We have provided further responses to the reviewer's specific comments below.

Again, thank you all for your time and input.

Associate Editor

Two experts in this subject area have now reviewed your manuscript. Both reviewers thought the extensive, long-term data presented in this ms have potential for contributing to our understanding of one or more basic questions of population dynamics. But, both reviewers also felt strongly that the importance of the study is less than clear because the results/statistics were not embedded in a broader biological narrative focused on a specific issue or question in population biology. In other words, the statistical findings (significant or otherwise) were not judged to be of sufficient value because they did not directly relate back to a specific question or hole in our knowledge (i.e, the narrative). To that end, the reviewers suggested that there are many possible narratives to choose from, and that the published literature could be better incorporated to develop a narrative. Finally, Reviewer 2 in particular thought the ms could be more strictly organized and clearer-particularly the presentation of results.

Reviewer(s)' Comments to Author:

Referee: 1

Comments to the Author(s)

I believe that density dependence is among the most important issues in population biology and the material in this paper could be important. But as presented we have not been persuaded that the results are novel and important. The authors need to review the current literature on density dependence and place their results in context. The debate regarding Sibly et al would be a fun place to begin but might not be the most relevant given the data presented here because we do not have sufficient data to rigorously evaluate the shape of the density-dependent function.

Line 37-38 This implies a concave density-dependent function (as per Fowler 1987) which seems reasonable to me, but Sibly et al. (2005, Science) challenge this premise.

We have substantially modified our introduction, including this section. Our intent, which is hopefully clearer now, was to highlight that most mammals today (marine mammals in particular) either exist at current carrying capacity or are in an extremely depleted and threatened state. There are only a few cases where there has been continuous, direct monitoring of reproductive metrics as a population has grown from near extinction to carrying capacity, and this is one of those cases.

Line 132 Using pups as index of density precludes reduced reproductive output as a density-dependent response. But seems to me we have adult numbers too.

In carefully considering both of the reviewers' comments, we realized that one point of conceptual confusion may be that the density metric being used (pup births) is intended primarily as a proxy for colony density during reproduction rather than overall population size. In this species, most adult females that are non-reproductive for a year are not present on the colony during the breeding season, but rather return to shore either earlier or later in their annual cycle. The approximation of pup births is based on female attendance during the breeding season only, not the number of reproductive females in the population. The intention in using pup births was to highlight the importance of colony density during lactation in modulating weaning mass. We chose to use pup births rather than adult female counts to be consistent with other publications on this species (e.g. Lowry et al. 2014 Aquatic Mammals, Le Boeuf et al. 2011 Aquatic Mammals), and since pup births is derived as a % of female attendance, the two metrics will have the same effect. The use of pup counts is common across pinniped species because they are often more reliably counted on colonies than adult females and these species typically only have single offspring.

We have made changes to the Methods sub-section on Population Data in an effort to clarify the intent of our proxy and some of the specifics relevant to this species. We have also modified the introduction to better highlight the density-dependent resource we are looking at – available beach space on the reproductive colony. Since this species has many reproductive colonies that all share a single (very large) foraging ground, a very different approach would be needed to examine foraging limitation as a result of population growth. We hope this clarifies several of the comments and concerns expressed by the two reviewers.

Line 183 I am surprised that these differences were statistically significant. Are the +/- numbers SEs?

All values reported were +/- st dev (Line 159)

Line 278 "Life-history theory is at the core of ecological and population dynamics" sorry, this is not true.

Removed/changed this text.

Line 282, you might review data on wolf recovery from Yellowstone.

A note has been added to acknowledge both wolf and humpback whale population recoveries.

Referee: 2

Comments to the Author(s)

1. Line 4: Word(s) seem(s) to be missing from "is primarily to ungulates"

Introduction was heavily modified, and this comment is no longer applicable.

2. Lines 32-34: I was a bit surprised that you didn't mention the probability of producing offspring.

Introduction was heavily modified, and this comment is no longer applicable.

3. Line 132: Did you have multiple functional forms for your proxy of abundance? That is, did you only consider linear relationships between abundance and the response variable(s)? I ask because one might imagine a non-linear relationship (e.g., theta-logistic like patterns).

In our models, we applied a smoother to our proxy of abundance to allow for a non-linear relationship between density and weaning mass. The edf in all cases was between 1 and 2.5 (smoothers shown in supplementary information).

In response to the following concerns regarding the modeling methods, we carefully reworked all of our modeling (and added an additional year of data) and substantially modified both the methods related to modeling and the presentation of our model results. We appreciate the reviewer's comments, as we believe this process has improved the clarity of our work.

4. Line 151: Did you constrain the flexibility of the fitted curve, e.g., using knots? If so, how? And, why or why not?

We have included more detail to the methods regarding the modeling process, including constraining the smoothers to 5 knots to avoid overfitting.

5. Lines 152-153: Did you use some threshold value for the correlation when deciding on including/excluding a variable? What was the range of values for the correlations and were any covariates eliminated? If so, how did you choose which to eliminate?

Yes, covariates with an absolute correlation value > 0.3 were not included in any models together. This applied to maternal mass and maternal age, which were both tested but were never included in the same model, and to the PDO which was highly correlated with all other ocean indices and was modelled separately.

6. Analysis methods in general: (1) did you have repeat observations on individual mothers in multiple years, and, (2) if so, how did you handle that in the analyses?

Yes, there were some repeat individual mothers, and we tested models that included MomID as a random effect (listed in Supplemental table of all covariates) but was not significant and therefore not included in final models. Explanatory text has been added.

7. Line 153: It would be helpful if you presented which response variables were analyzed and what competing models were evaluated for each (or note that the same models were used for each). You note that model selection was done, but it's not clear to me from the presentation what the list of competing models was or how models were constructed. Did you evaluate all-possible models or just models with different versions of the indices of ocean conditions? What

of other covariates and how they were in/out of competing models? Finally, did you have any specific predictions regarding the covariates that might be useful to describe here?

This methods section has been substantially revised and hopefully clarifies what was done to address the questions presented here.

8. Line 182: “higher than in the” instead of “higher than the”

Corrected.

9. Line 183: I don't follow the phrase that reads, “ 127.5 ± 23.5 versus 125.1 ± 21.7 respectively; $p=0.0024$ ”. First, what of the biological significance of the difference: it might be helpful to consider that topic in introduction or methods. And, it would help to know what the value provided after the “+” sign represents, e.g., if it's SD, we might want to have the SE of the mean and the estimated difference along with a CI on the difference. Further, the values after the “+” sign are quite large relative to the estimated, which further indicates a need to. Understand what values are being provided after the “+” sign. Finally, this paper provides a mix of significance testing (p-values presented here) and information-theoretic results (AICc differences), which is non-standard and a bit confusing. It might be helpful to pick an analysis paradigm and use it throughout.

The biological significance is that greater weaning mass mean a higher probability of surviving to adulthood. While this concept was implied or assuming in our introduction, it was not explicitly stated until the discussion section of the paper. We added language to the introduction to explicitly address this concept from the start in addition to clarifying the concept in the discussion.

All values reported were +/- st dev (Line 159), we chose to report sd because it gives a clearer picture of the variance in the data, which is obscured in standard error, especially given the large sample sizes here.

We felt that both significance and information-theoretic results had a role to play in first assessing whether or not differences were present (e.g. are weaning masses lower/higher in some years) and then evaluating the multi-causal mechanisms underlying that variability (importance of maternal age, environment, etc.). Mixing the two approaches may be unusual, but it is not unprecedented, and agreement between the two methods adds confidence to our findings (Stephens, et al. 2005 Journal of Applied Ecology).

10. Lines 193-194: Here the paper is relying on p-values it seems rather than, as stated on lines 153-154, “the Akaike Information Criterion (AICc)”. Whatever approach is chosen should be justified and used consistently in my opinion.

As stated at the start of the “Quantitative Analysis” section, ANOVA were used to test for differences between years prior to an examination of causative variables using information-theoretic methods. In an effort to better clarify our use of the two methods, we substantially modified the Quantitative Analysis subsection of the methods, especially the opening paragraph and all of the text regarding our modelling efforts.

11. Lines 192-196: Here (and throughout much of the results), it is not possible to assess the

magnitude of the differences from the main text. Yet, I think that many readers will be more interested in those features of the data than in values such as the % of deviance, F statistics or P-values. It would be helpful to bring more of the biological results into the text to accompany the statistical results.

The results section (including modeling results table) has been heavily modified to address these concerns.

12. Line 200: By “significantly”, are you referring to biological or statistical significance or both? IT would be useful to clarify and justify the choice.

Significantly here meant statistical significance, the text has been clarified.

13. Lines 217-220: Same comments as made above regarding text on lines 192-196. The results section (including modeling results table) has been heavily modified to address these concerns.

14. Line 236: Same comment as made above regarding text on line 200.

Significantly here meant statistical significance, the text has been clarified.

16. Results in general: I found it difficult to evaluate which of the covariates were most important in explaining variation in your response variable(s) and suggest providing more concrete information on the topic in Results.

The results section (including modeling results table) has been heavily modified to address these concerns.

17. Literature Cited: There are other papers on marine mammals in particular that relate to some of the topics covered in this paper and that could be added, e.g., Bowen et al. 2015. Ecology and Evolution: 5(7): 1412– 1424.

We agree with the reviewer that there are numerous additional studies that could be cited throughout. Due to concerns about length, we had narrowed our citation list from ~90 to the 60 we considered most essential to the points being made; we did add the Bowen paper mentioned here to the discussion about the importance of weaning mass.

18. Table 1: By “Year (RE)”, are you indicating that year was treated as a random effect? This should be explained here and in Methods regarding the how and why of including it this way.

In reworking the modeling, this piece is no longer included.

19. Table 2: What values are being provided after the “+/-” sign?

All values reported were +/- st dev (Line 159), will add clarification to table legend

20. Figure 1: It appears that both response variables hit what might be considered a threshold well before the point in the study at which you consider the density to have transitioned from low to high. Is this adequately discussed? What are the implications?

In considering this comment, we have chosen to simply use colony density (# of pup births) as a covariate in our models and have not split the data between time periods anymore. However, to illustrate/highlight the effect of colony density, we compared years of low density (pup births <1200) to years of high density (pup births > 1900), which is more biologically meaningful than a simple temporal split (see Table 1 and Figure).

21. Figures 2 and 3: Confidence bands to provide measures of uncertainty are lacking but would be useful.

Confidence bands are now included in all figures.

Appendix B

We would like to thank the editor and reviewer for their time and for their contributions to the refinement of this manuscript. We have striven to address all the concerns and questions posed below. For clarity, we've left the full text of the reviewer's comments (in black) and have responded one section at a time (in blue).

Reviewer #3:

Chiefly, while I agree with the authors that pup births should be used as the density metric, I have concerns over the scale of what is defined as 'the population' and thus the 'density' thereof. From the outset, it is clear that an increase in absolute numbers of NES pups relates strongly with a decline in weaning mass. However, this is not the same as saying an increased density of individuals on land is cause for observed decrease in weaning mass, because this is a question of scale of distribution and spacing of the individuals on land. This in turn is heavily dependent upon the behavior of the species. Specifically, how 'the population' relates to effects of density at the individual level, which would ultimately translate to weaning mass, given that age and experience of individual mothers (and thus their behavior and physical location in colonies on land) is provided as essential explanation for the density effect (Lines 207-214).

The terms 'population' and 'colony' are used vaguely in the manuscript. For those not familiar with the Año Nuevo NES assemblage it is important to clearly explain if the NES here are assembled in one large continuous population more synonymous with colony (likely not), or if as suspected, the population here is in fact a 'colony' of several harems of varying sizes dotted along the coastline. Are all harems distinct and separated from each other or are some margins vague and harems merge into each other? These questions are critical to address because I am concerned that a confounding argument of scale of study may be associated therewith.

Clearly an increasing population will mean more individual animals within a defined area as a whole, but does this mean that harems are in closer proximity to each other, or perhaps merge? Does an increase in population size mean greater congestion (i.e. density) within harems? Does it mean the disappearance of small harems because all harems grow beyond a specific size threshold? Ultimately, are a larger number of NES merely spread over a larger space at a constant density or is the density of harems increased?

These are vital considerations to be sure that what the authors are testing here as terrestrial population density dependence is indeed causative to the responses, i.e. changes in weaning mass. Or are increased numbers of animals in the population resulting in a density dependent effect at foraging grounds at sea (not the focus of this study). Or are we seeing a response related to how increased numbers of animals are changing the dynamics of terrestrial harems, which we know have different consequences for mothers of different ages. Clearly, density is integral but establishing at what scale the effect is would be a far more valuable addition to our knowledge than what is currently proposed in the manuscript I think.

We appreciate the reviewer's questions here, as these are important points to clarify. We have added text to the Methods section (lines 72-73 and 127-129) and included additional figures (map and drone imagery) and census data to the supplemental materials to help clarify that an increase in population does translate to an increase in terrestrial density at this site.

The mainland portion of Año Nuevo is divided into two large areas (referred to as North Point and South Point) which, at peak colony attendance, are mostly continuous groups of animals. The bulk of the population (~80%) breeds on these continuous beaches. These areas are

further divided into small defined geographic areas (illustrated in the map we've added to the SI) that have been used for census counts and to identify the location of known individuals throughout the years of study. Interestingly, the animals have not moved outward into additional areas as the population has grown, simply increased their numbers within established areas. We have added some census data from 1985-2020 to the SI showing counts of adult females at the two primary breeding areas at South Point to illustrate the increase in population within the smaller, constrained geographic areas (meaning an increase in congestion/density within the harems as opposed to a constant density spread over a larger area). Additionally, we have included drone imagery of the largest breeding area of the colony to illustrate the distribution/continuity of animals we describe here. For these areas, particularly the Año Point Area, it is difficult if not impossible to identify harem boundaries among the 500-800 breeding females, although numerous alpha and beta males are present.

While at-sea density effects are certainly an important consideration, the decline in wean mass and population plateau seen at Año Nuevo occurred during the 1990s, at which time the overall NES population was rapidly growing (20k births/year in 1990 to 30k in 2000 and 40k in 2010 – Lowry et al. 2014). Tracking data from various colonies in California and Mexico show that these animals share a common foraging ground (Robinson et al. 2012, Kienle 2019), reducing the likelihood that at-sea density effects were the primary cause of a decline in wean mass while the larger population was still growing rapidly. Additionally, our adult female mass measurements do not show a temporal trend of decreasing arrival mass as the species population has continued to increase (measurements from 1991-2020).

We have additionally reconsidered our use of the terms “population” and “colony” throughout the manuscript to ensure that our intended meaning is clear.

S8 & S10 seem to be valuable additions to the main manuscript rather than being deferred to the Suppl Mats.

We agree that these figures are both valuable. Due to constraints on length, we've chosen to add S8 to the main manuscript and have left S10 as a supplemental figure.

Also, the means illustrated in S10 as compared to the total population combined decline in weanling masses over time (S8), seems to me to indicate that the age distribution/composition of the adult population in harems has changed over time.

Colony density as a function of population increase will be heavily dependent on available space for harems, and will likely vary between different harems? Evidence from SES suggest that young females are more regularly associated with small harems, while older more experienced females associate in larger harems with varying consequences for their pups (Pistorius et al. 2001 Polar Biol; McMahon & Bradshaw 2004 Behav Soc Biol; Postma et al. 2013 Polar Biol). If NES do the same the potential caveat here is that harems of different densities present differing female age distributions, which will confound the conclusion that density-dependence at the population level is driving the observed breeding output changes.

A past study on NES harem density and maternal age at Año Nuevo Island (Reiter et al. 1981) found that older females tended to occupy the center, more prime locations within a harem,

while younger females tended to be forced to the outskirts. Older females were more successful at weaning healthy pups than young females across the board, but that difference was much greater in larger harems. However, they did not see young females favoring smaller harems, despite the apparent fitness benefit. In addition, the more or less continuous nature of the bulk of the harems at Año Nuevo negates this particular issue as there are very few harems at the colony that could be identified as small or low density, nor have there been since early in the establishment of the mainland population.

I feel that the manuscript is not streamlined in its present form, nor do I think it is adequately contextualized within existing literature. A host of highly relevant papers, some asking/answering several of the same questions (albeit for SES) seem to have been missed. I provide some added examples in the specific comments below, but from the outset Oosthuizen et al. (2015 – Ecosphere), seems to be a glaring omission. That paper; “Decomposing the variance in southern elephant seal weaning mass: partitioning environmental signals and maternal effects”, provides key context to many aspects addressed here (e.g. their figure 4 looks very similar to your figure 2).

We appreciate the reviewer bringing this particular paper to our attention as this was a notable oversight and we have added several of the Oosthuizen papers to our manuscript. We recognize that there are numerous additional references that could be included but are constrained by space and have striven to retain only the citations that were most relevant. We have gone back through all of the references used in the introduction and discussion to reconsider if they were the optimal pieces to include and have made some changes in addition to the inclusion of the Oosthuizen pieces.

We also went through in an effort to tighten, streamline, and clarify the text, particularly in the introduction and discussion.

Specific comments:

Title: ‘Reproductive output’ would be better than ‘reproductive success’. “LRO is a slightly more general concept than lifetime reproductive success (Clutton-Brock 1988) because it can accommodate many different operational definitions of reproduction, as are often encountered in ecological and demographic analysis.” - <https://link.springer.com/article/10.1007/s12080-017-0335-2#ref-CR18>

Changed

Line 10: This seems to be a strong statement for what appears to be fairly limited male-biased allocation of resources to offspring?

Removed

Line 10-11: Just a note - while this may be novel for NES, it is not for the southern species (see Oosthuizen et al. 2015 – Ecosphere).

Noted

Line 59: incomplete sentence.

Corrected

Line 61, Hypothesis 2: Refer to general comments. Your design is not necessarily testing a higher density population, but rather a larger (in number) population, unless more detail is provided that motivates that the population has both grown and increased in density at the terrestrial site.

See earlier explanation regarding terrestrial density.

Line 87: if they were measured then why did they need to be corrected? Unclear. I'm assuming because measurements were not always taken at times in the life cycle needed for later analyses. State your reasoning here.

Added a clarification to the previous paragraph that pups and females are not always measured at the same time post-partum (line 86). Additionally clarified the motivation for mass corrections at what was line 87 (now line 90).

Line 139-140: see Oosthuizen et al. 2015 Ecosphere, Fig 5. Individual heterogeneity in the context of the general comments related to individual age related behaviour of female at the terrestrial breeding site may in fact be a critical component of what happened demographically to this population as it grew.

We agree that individual heterogeneity is likely a critical component of reproductive output – other studies on NES have illustrated the importance of variation of overall maternal quality, highlighting this idea of super-moms (Le Boeuf et al 2019), although it was not intended to be the focus of this particular study. In carefully reassessing our data and models to address this particular comment, we found some additional wean mass data that had known identity and age moms, including more repeat moms. We have incorporated those data and re-run all of the models and retained MomID as a random effect in models 1.a-h (the data set for models 2.a-d did not include repeat measures from the same mom). These data have strengthened the fit of our models but have not changed the results or story of our paper. We have updated Table 1 and all of the supplemental figures showing smoothing curves to reflect the new models, and address the topic of individual heterogeneity in lines 230-233.

Line 144: would Chlorophyll-a be a useful variable to include?

We recognize that chl-a is one of several meaningful proxy for ocean productivity and overall conditions. We chose to use some of the cyclical oceanic indices that dominate the north Pacific (ENSO, PDO, and NOI) because these indices encapsulate basin-scale physical oscillations that result in changes in circulation patterns. This in turn governs not only some of the variation seen in gross productivity, but also changes in species composition, locations, and intensity of aggregations for both phytoplankton and lower trophic level consumers (e.g. Mantua and Hare 2002, Chiba et al. 2012, Siswanto et al. 2016, Keister et al. 2010). We felt that these indices adequately captured the ecological variation in the NE Pacific at a level that was appropriate to the questions being asked.

Lines 182-185: Might, as stated, the crux in the argument of declining weaning mass over time not be with how the breeding season terrestrial behaviour of older females and/or the composition of females in different harems has changed over time?

A brief look at the maternal age data available indicate that, if anything, mean age of the reproductive females has increased as the population approached its peak size, which should favor larger pups at weaning not a decrease in mean weaning mass. But that increase in age does not account for the substantial year to year variation in tagging effort, loss rates of different tag types, and other factors that could lead to artificially inflated representation of certain cohorts. As previously mentioned, most of the reproductive harems cannot be clearly divided from others, so understanding harem age structure for up to 70% of the adult females is not feasible in the way harems have traditionally been considered. While we believe that both of these questions are worth additional consideration but are beyond the scope of the present study.

Line 240-241: speculative, suggest deletion.

Deleted.

Line 242: Indeed. Several pertinent references from the work on SES at Marion Island may provide added insight on this specific topic here:

Addressed above – additional references have been included and all introduction and discussion references were reconsidered.

- Pistorius et al. 2001. Pup mortality in southern elephant seals at Marion Island. *Polar Biol* 24:828–831
- Postma et al. 2013. Age-related reproductive variation in a wild marine mammal population. *Polar Biology* 36:719–729.
- Oosthuizen et al. 2018. Phenotypic selection and covariation in the life-history traits of elephant seals: Heavier offspring gain a double selective advantage. *Oikos* 127:875–889
- Oosthuizen et al. 2019. Individual heterogeneity in life-history trade-offs with age at first reproduction in capital breeding elephant seals. *Population Ecology* 61:421–435.
- Oosthuizen et al. 2021. Positive early-late life-history trait correlations in elephant seals. *Ecology* 102: e03288

Line 246: 'higher' may be better replaced with 'greater'?

Changed

Line 247-248: again see above Oosthuizen et al. 2018, 2019, 2021 references.

Addressed above.

Line 249-260: this paragraph is valuable but feels a little 'added on' here as an afterthought almost. Attempt to link better.

Modified lines 256-259 to better tie to the previous paragraph.

Lines 286-289: based on the general comments above this line needs some reconsideration.

Given the clarifications with regard to terrestrial density at this colony, we feel that this statement can remain as written.

Editor Comments:

I would add to this that I would like to see a few more lines justifying the hypotheses in the introduction, particularly 2 and 3.

Thank you for bringing this concern to our attention - we have added a few lines to the introduction to better set up and give context for these hypotheses (lines 45-50).